# Fuzzy-Based Efficient Healthcare Data Collection and Analysis Mechanism Using Edge Nodes in the IoMT

**DOI:** 10.3390/s23187799

**Published:** 2023-09-11

**Authors:** Muhammad Nafees Ulfat Khan, Zhiling Tang, Weiping Cao, Yawar Abbas Abid, Wanghua Pan, Ata Ullah

**Affiliations:** 1School of Information and Communication Engineering, Guilin University of Electronic Technology, Guilin 541004, China; nafees.ulfat@mails.guet.edu.cn; 2Guangxi Key Laboratory of Wireless Broadband Communication and Signal Processing, School of Information and Communication, Guilin University of Electronic Technology, Guilin 541004, China; weipingc@guet.edu.cn (W.C.); pwh2008_79@163.com (W.P.); 3School of Computers and Cyberspace Security, Guilin University of Electronic Technology, Guilin 541004, China; yawar.abid@cuisahiwal.edu.pk; 4Department of Computers Science, COMSATS University Islamabad, Sahiwal Campus, Sahiwal 57000, Pakistan; 5Department of Computer Science, National University of Modern Languages (NUML), Islamabad 44000, Pakistan; aullah@numl.edu.pk

**Keywords:** Internet of Things (IoT), fuzzy logic, data aggregation, healthcare, FIS, member functions

## Abstract

The Internet of Things (IoT) is an advanced technology that comprises numerous devices with carrying sensors to collect, send, and receive data. Due to its vast popularity and efficiency, it is employed in collecting crucial data for the health sector. As the sensors generate huge amounts of data, it is better for the data to be aggregated before being transmitting the data further. These sensors generate redundant data frequently and transmit the same values again and again unless there is no variation in the data. The base scheme has no mechanism to comprehend duplicate data. This problem has a negative effect on the performance of heterogeneous networks.It increases energy consumption; and requires high control overhead, and additional transmission slots are required to send data. To address the above-mentioned challenges posed by duplicate data in the IoT-based health sector, this paper presents a fuzzy data aggregation system (FDAS) that aggregates data proficiently and reduces the same range of normal data sizes to increase network performance and decrease energy consumption. The appropriate parent node is selected by implementing fuzzy logic, considering important input parameters that are crucial from the parent node selection perspective and share Boolean digit 0 for the redundant values to store in a repository for future use. This increases the network lifespan by reducing the energy consumption of sensors in heterogeneous environments. Therefore, when the complexity of the environment surges, the efficiency of FDAS remains stable. The performance of the proposed scheme has been validated using the network simulator and compared with base schemes. According to the findings, the proposed technique (FDAS) dominates in terms of reducing energy consumption in both phases, achieves better aggregation, reduces control overhead, and requires the fewest transmission slots.

## 1. Introduction

The Internet of Things (IoT) is made up of smart devices that can communicate with each other by exchanging information [1]. These devices include multiple intelligent sensory elements and wearable smart devices, which are crucial for the development of the IoT [2]. The IoT has become an integral part of various fields such as healthcare, mining, buildings, cities, agriculture, transportation, industries, smart homes [3], smart surveillance [4,5], and automated systems [6]. Smart medical devices in healthcare can connect people and objects, making life easier and more convenient [7]. The Internet of Medical Things (IoMT) has become a crucial component of healthcare, offering intelligent services by collecting different types of data and transmitting them to cloud-based repositories [8,9]. The integration of the IoMT into smart healthcare has enabled seamless connectivity. As a result, developing an environmentally sustainable [10] solution to address the multiple challenges faced by the latest IoT-based smart healthcare strategies [11,12] is critical. Medical devices enable remote monitoring of patients, resulting in improved quality and efficiency of medical treatment.

Health information is collected from patients’ sensor devices and then transmitted to smart collectors in a secure manner in both normal and emergency situations [13]. Miniaturized devices play a crucial role in healthcare data collection, where security is quite critical [14] for efficient authentication  [15]. Cyber–physical systems (CPSs) are utilized in social services, particularly in healthcare applications, as cost-effective solutions [16]. In health monitoring, the health-related data of patients are transmitted to the cyber world to allow for real-time processing and analysis of vast amounts of data [17]. In this scenario, enhanced computing frameworks are necessary to dynamically integrate both real-world and cyber aspects of medical cyber–physical systems [18]. IoT-enabled medical networks can manage complex communication to handle the processing of many users [19].

Fog computing architectures act as a middle layer between cloud servers and end users, providing data computation, storage services, and networking capabilities. The term “FoG server” was first introduced by Cisco [20]. The smart healthcare architecture [21] enables monitoring devices to communicate with patients and transmit data to a server remotely [22,23]. At the edge of the network, the smart healthcare architecture processes large amounts of data generated by numerous devices to reduce bandwidth and energy consumption. This reduces overhead on the cloud server and balances the load among multiple local fog nodes by integrating fog and the IoT [24]. Fog nodes may make intelligent decisions in emergency situations to efficiently handle critical health issues [25] with smart collector nodes [26]. The combination of fog computing and cloud computing can be an appropriate solution to overcome challenges in the IoT and healthcare systems [27,28].

Data aggregation is a critical technique used in the IoMT to collect health parameters from sensing devices and transmit them in a collective manner to reduce the transmission cost [29]. Furthermore, to optimize the data aggregation processes, mobile devices have been introduced as collector nodes [30]. For better collection and analysis of the data, fuzzy logic is employed. An efficient fuzzy-based healthcare data collection analysis mechanism is an approach that utilizes FL to analyze data. These data are gathered from wearable sensors, implanted devices, and some other resources. FL uses a mathematical framework to manage patient data and, in return, provides decisive information to healthcare professionals. This mechanism is needed in the healthcare sector because FL efficiently tackles huge, sophisticated, and varying patient data. It examines changes in patients’ data readings over time, assists doctors in making rational decisions timely to avoid health complications later on, and is cost-effective. These benefits cannot be attained with conventional methods. Robust security measures are demanded in IoMT systems to preserve patient data privacy and integrity [31,32].

Ensuring secure and privacy-preserved aggregated data is a crucial and mandatory aspect of both edge–node devices and fog nodes [33]. To maintain both data integrity [34] and to authenticate edge devices, authentication is also required using encryption-based measures  [35]. Recent surveys add to the body of knowledge on aggregating healthcare data using IoT-based sensing devices [24,36] and the applications of fog computing [22,37]. However, these surveys do not address the security measures needed during the transmission of aggregated data. On the other hand, [27,35] considered security measures but did not extensively explore IoT scenarios. In [38,39,40], secure data collection and aggregation scenarios were explored but fog-assisted approaches were not considered.

The proposed technique, called the fuzzy data aggregation system (FDAS), maintains a high data aggregation rate in heterogeneous environments, even when the number of attributions increases. After studying the literature on AI-based data aggregation, the current problem was identified. The proposed methodology emphasizes effective data aggregation and the elimination of duplicate values to boost network performance and decrease energy consumption. The aggregated data from various sensors are checked for duplicate readings before being sent to the central server. To check the effectiveness of the proposed scheme, a simulation was conducted using NS-2.35. The results were compared with those of some previous robust schemes in terms of some crucial metrics. The proposed scheme significantly reduces data size and reduces communication costs, making it an appropriate choice for use in the healthcare sector.

The main contributions of this paper are as follows:We studied the most relevant literature on data aggregation using artificial intelligence techniques.We developed the FDAS, a scheme that uses fuzzy logic to select a suitable parent node for each child node in a heterogeneous environment.We developed a detailed mechanism for dealing with in-range normal data by sending Boolean digit zero to reduce the size and transmission of duplicate messages.We conducted simulations and compared the results of the FDAS with those of some prominent schemes, including FAJIT, DQN-FATOA, DICA, and DICA_EXTENSION.

The rest of the paper is organized as follows: Section 2 explores the literature related to fuzzy-based data aggregation schemes. Section 3 explores the system model and the problem statement. Section 4 presents the proposed fuzzy-based solution for data sharing, and Section 5 presents the results and discussion. Finally, Section 6 concludes our paper and highlights future work.

## 2. Related Works

This section describes the literature on data aggregation schemes and the role of AI in enhancing the overall mechanism. The data are gathered from many sensor nodes that collect the data and share the aggregated data with the base station (BS) and central repositories in the cloud in IoMT and wireless sensor networks (WSNs). Data aggregation aims to efficiently transmit large amounts of data to BS to increase the network’s lifespan. For efficient data aggregation, artificial-intelligence-based schemes have been explored. The AI techniques described in this part have been used in many of the most recent research in this field.

### 2.1. Data Aggregation Using Artificial Intelligence Techniques

Kulkarni et al. highlighted the use of a computational intelligence (CI)-based algorithm. These CI-based algorithms deal efficiently with dynamic environment sand limited node energy. As CI-based solutions are not perfect under some conditions, artificial neural networks (ANNs), genetic algorithms (GAs), and particle swarm optimization (PSA) can be used [41]. In [42], swarm-intelligence-based schemes were elaborated that have improved network lifespan and use energy efficiently. Chen et al. presented the data aggregation ACO algorithm (DAACA) to utilize network energy efficiently, which is based on the ant colony optimization algorithm approach. The DAACA has features of both local and global pheromone methods. The different versions of DAACA assist in reducing computational and communicational overhead and increasing node life. The advantage of this scheme is that it features fault tolerance, low complexity, and higher flexibility [43].

Imitating the dynamic nature of a river, intelligent water drops (IWDs) are used in WSNs, where every drop shows a solution. This method is deployed after amendments in tree-based data aggregation to obtain optimal solutions [44]. To economically use energy in WSNs, an ant-colony-based optimization scheme called the minimum incremental dissemination tree (AMIDT) was introduced. This scheme consists of two main stages: in the first stage, an online tree is formed; during the second phase, a path- and reference-based heuristic is applied. AIMDT has reduced cost and energy utilization compared with existing schemes [45]. For different optimization issues, the ABC algorithm outperforms others in cases where nodes do not directly transmit data to the aggregator node or BS. Knowing the best traveling path of a mobile robot is important for reducing energy consumption. For this issue, the ABC algorithm performs better than greedy algorithms. The positive point of this scheme is that the retrieved results are very stable but cannot be implemented in the case of multiple robots [46]. As the number of sensors increases, the data they produce also increase, which is problematic for data mining techniques. To overcome this problem, a decentralized scheme using a WSN neural network was presented. A data technique was trained using data gathered from nodes in the network. The system performs well and was used for higher WSNs [47]. An optimum cluster was chosen using the shuffled frog algorithm. It showed better efficacy, fast searching, and optimal energy usage. CHs were selected based on the residual energy of WSN nodes [48]. The PSO method was used to aggregate data from complex and large networks. These methods assist in finding a better transmission path between nodes that lessens hop distances but increase hop counts. This method has lower energy consumption and is suitable for the dynamic nature of the environment [49].

The F-LEACH scheme was presented to extend the network life span and economical energy utilization. Selecting an appropriate cluster head improves network performance. For this purpose, the FIS function was used to determine cluster heads in the network. FIS considers the distance of nodes from the base station and residual energy level; a node with a low distance from the BS and high energy has a high probability of being selected as the cluster head. If N nodes are in the network and the total clusters are K, there are approximately N/K clusters in the network. One of them is a CH node, while others are ordinary nodes. The proposed scheme has fewer dead nodes, improved residual energy, and improved work performance up to 5–20% [50].

Abid et al. designed a scheme that deals with the problem of efficient data aggregation and transmission in time-constrained wireless sensor networks. For the aggregation of data, multilevel clustering was used, in which a structure-free approach was considered. In this clustering, nodes are divided into large clusters during first-phase clustering. Each group has one cluster head (CH) collecting member data. The CH node needs more energy to work efficiently, so it is important to check the energy level of nodes from time to time and select the CH node intelligently. For this purpose, the event-driven cluster head election (EDC) algorithm was used in which node residual energy is checked against threshold energy E0 and checking the CF bit. If the current CH fulfills both conditions, no change occurs. Otherwise, a node near the present CH is selected as the next CH to reduce energy wastage. The proposed scheme was simulated using NS-2, and the results proved that the proposed method achieved better aggregation gain and lower delay. The drawback of the process is that if the energy level of primary CH dissipates, the communication of the whole network is affected [51]. Singh et al. proposed an energy-efficient scheme to overcome the probability of transmitting redundant data and balanced energy utilization. For clustering, four main parameters were considered in the fitness function: the energy level of the node, density near CH, Euclidean distance of the CH and the sensor nodes calculated usinf Equation (Equation 1) [52], and distance from CH to BS.
(1)$$\mu_{1}=\frac{\sum_{k=1}^{K}d\left(n_{i}CH_{ek}\right)}{\left|C_{\varepsilon ,k}\right|}$$
where $\mu_{1}$ denotes the maximum Euclidean distance among nodes and the respective CH, while $C_{\mathrm{bejk}\;}$ represents the total nodes lying in the communication range of egg e of a cluster named *k*. to calculate the total amount of energy of nodes lies in a network; Equation (Equation 2) [52] is used.
(2)$$\mu_{2}=\frac{\sum_{k=1}^{N}E\left(N_{l}\right)}{\sum_{k=1}^{K}E\left({\mathrm{CH}}_{e,k}\right)}$$

Using a fitness function, 20% of the nodes are considered for CH, and after calculating their cost, the best host nest is obtained. $\mu_{1}$ denotes the maximum Euclidean distance among nodes and their respective CH; Equation (Equation 3) [52] is used to calculate cost *C*. The value of λ is considered as 0.5. The lower value obtained by $\mu_{1}$ and $\mu_{2}$ assists in reducing the intracluster distance, which ultimately helps with the selection of ideal CH.
(3)$$C=\lambda *\mu_{1}+\left(1-\lambda \right)*\mu_{2}$$

Vasim et al. presented a scheme emphasizing prolonging network life while efficiently consuming energy. The proposed method consists of four important phases. In phase 1, node mobility is monitored using the distance formula. Mobility dissipates energy quickly, so the node having low mobility is selected to minimize utilization. For determining CH, the AE-LEACH algorithm is implemented in the next phase, which uses residual energy, and distances from BS are used. A threshold value is calculated for all the nodes that lie in the range of zero or one. A certain node works as a head if the computed value exceeds the capacity. The CH broadcasts messages in the network; all nodes that reply to that message become cluster group members. Member nodes transmit data to the CH, which are sent to the BS after performing aggregations. The particle filter algorithm estimates the targets’ next state in the third phase. In the last step, the Gini index is used to check the even consumption of energy level. NS-2 evaluates the proposed scheme and provides better results in network lifetime, energy utilization, and residual energy. The benefit of this scheme is that the CH is not fixed and changes frequently, reducing the burden on the node playing the role of CH. The drawback of this technique is that the case of malicious nodes is not handled efficiently [53].

Amutha presented a hybrid scheme that caters to both cases when the sink node is static and moveable. In both scenarios, the CH is selected by checking some circumstances, including residual energy, node density, distance factor, and node centrality. The node centrality is a new parameter that has yet to be included in previous studies, which leads to selecting the optimal CH; the members a short distance from the CH and high energy are clustered under that head. For the static case, static sink nodes (SSNs) are placed in left, right, and center in an area of interest (AOI) to check the CH’s energy level and effective path. In the second case, the mobile sink (MS) node moves randomly to collect data. The mobility model was introduced to resolve the hot-spot problem in a later approach. The simulation was conducted using NS-2; for different metrics, it showed better energy consumption and data delivery, low delay, and higher throughput than former schemes. The advantage of this scheme is that in the case of a static sink node, all possible directions for calculating the optimal path to disseminate aggregated data to SN are considered. The problem in the case of MS is that it works well only for short-range communication but not long-range communication  [54].

Yan et al. proposed using game theory to cluster sensor nodes and reduce energy consumption in wireless sensor networks (WSNs). Each sensor node is viewed as a player node and uses its current state (active or passive) to cluster them. GEC introduces the transition of active nodes into a sleeping state and vice versa when required and introduces penalty principles to control energy violations caused by selfish or greedy nodes in the network. These penalties aim to reduce the energy consumption of communicating nodes in the network [55]. The benefit of this scheme is that it provides reliable performance even in harsh environments. The drawback of this scheme is that it is not suitable for heterogeneous WSNs. To reduce energy consumption in a heterogeneous WSN environment, Bhushan et al. presented a fuzzy-attribute-based joint integrated scheduling and tree formation technique, which intelligently selects different parents for different types of nodes. Two phases were defined: control phase and data phase. In the control phase, nodes choose lots and parents, and they have information about the number and type of data packets and the type of packet that the node itself has generated. In the second phase, nodes are created, and data are transmitted. For each outgoing data packet, a different parent can be chosen per its type. To fuzzify the system, min–max normalization is used to scale the weights. These weights act as membership functions, having small consequences with a straight connection to a node having a high probability of being selected as the parent for data aggregation and transmission to an SN. For packet t, a neighbor of type t is chosen; if it is present, data are forwarded to it. Otherwise, having the node’s neighbor of type t, data are forwarded to it. In case no such scenario occurs, the node with the fewest dynamic nodes is considered the parent node. For checking the performance of the proposed schemed FAJIT, a simulation was performed and compared with DICA and DICA_EXTENSION under different metrics. The results showed that FAJIT performs better than the previous two schemes. The benefit of this scheme is that it is considered a heterogeneous environment for sustaining energy levels. The drawback of the method is that complications increase when deploying it for large networks [40].

In [56], the author proposed a scheme for collecting data in which sensors are deployed. After that, clustering is performed, and data are saved into a repository. To cluster uncategorized data, the K-means clustering method is used. The proposed scheme, fuzzy-logic-based data aggregation (FLDA), uses fuzzy logic. Afterward, fuzzification and defuzzification are applied to obtain a discrete output to aggregate the most appropriate data, decreasing the volume of duplicate data. The proposed scheme was simulated in a MATLAB MATLAB 2018b environment, and the results showed that FLDA achieved better data persistency, higher network lifetime, and reduced energy utilization. The drawback of the scheme is that it works better under specified assumptions that cannot always be achieved as the density of nodes increases.

By considering the hot-spot or energy hole issue in WSNs, Ssert et al. presented a technique that uses energy economically. The proposed method, two-tier distributed fuzzy-logic-based protocol (TTDFP), is categorized into two tiers. In the primary tier, fuzzy logic is used to select CHs for unequal clustering via the process of the probabilistic model. In the second tier, the routing path is chosen. The cluster head generates the Min value, which is assigned as one. The fuzzy logic is applied if the min value increases and becomes >1. The proposed scheme was simulated with MATLAB, and the results showed that TTDFP achieved a higher ratio of the alive node’s residual energy until the last round. The advantage of this proposed distributed scheme is that the hot-spot problem is resolved. It balances and economically consumes power, which ultimately increases network lifespan. The limitation of the proposed method is that no procedure is initialized to lessen the load on the CH [57].

The grid clustering method was introduced to efficiently aggregate data from nodes by utilizing an economical amount of energy. The whole area is divided into grids, and a CH is selected for each. The entire network acts as an environment, and the CH works as an agent. The nodes with a short distance and good link quality are more likely to be selected as aggregator nodes. Afterward, the sink node is placed occasionally in areas where a low amount of energy is used. For this purpose, the fruit fly optimization algorithm is used. MATLAB was used for simulation and generating results considering important metrics such as PLR, energy utilization, and throughput. The benefit of this scheme is that it reduces latency and enhances node lifetime by minimizing energy consumption. The drawback of the method is that complications arise, and performance degrades in the case of dense WSN [58]. To maintain efficient data aggregation in heterogeneous networks, the scheme uses fuzzy logic scheduling, which takes two inputs: the residual energy of a node and the overlapping range of a node and the nodes in its vicinity. The output of the system is a suitable data rate. In the second stage, the binary tree of a CH is created, and the BS is responsible for implementing the dragonfly algorithm to create an aggregated tree. Afterward, high-priority nodes are placed as left and right children. If two nodes have the same priority level, in this case, the node with low priority is considered first. When the tree is completed, it is evaluated based on a fitness function. The fitness function is calculated using Equation (Equation 4)  [23].
(4)$$F=\sum_{D=1}^{\log n}\frac{1}{D}\left(\sum_{i=1}^{c}\frac{1}{\frac{d_{i}}{a_{\max}}}\right)+\left(\frac{\;\mathrm{Residual}{\;}_{\mathrm{energy}\;}}{E_{\max}}\right)+\left(\frac{1}{\frac{n_{\mathrm{maxster}\;}}{n_{\max imam}}}\right)$$

This equation checks factors including distance to the sink node, residual energy of CH, and the number of CMs. Then, it evaluates the constructed tree based on the abovementioned terms and selects the best tree. *D* shows the depth of the present aggregated tree, *n* is the total CH in the network, *c* is the CH in a current tree, *d* is the distance between CH and $\mathrm{SN},G_{5}$ is the maximum distance between CH and SN, $E_{\max\;}$ is the ultimate energy level of CHs, the $n_{-}\mathrm{cluster}$ is the total CM in a cluster i, and $n_{\mathrm{maximum}\;}$ shows the entire CM in the group. The proposed scheme was simulated using NS, performing better than existing schemes. The benefit of this scheme is that it provides better data packet transmission because of a suitable aggregation tree that was not considered in previous methods. The drawback of this technique is that the CH has to bear a higher load that eventually consumes more energy [59]. This protocol introduces a multipath for aggregated data to transmit crucial healthcare data to medical servers (MS) and reduce delay. The incoming data are divided into normal and emergency data. When biosensors send readings, if the generated data are beyond the normal range, it sets the threshold value as one. This sort of data are termed emergency data and are transmitted to cluster heads from the best routes with the lowest congestion rate. On the other hand, normal data are transferred from ordinary routes. Fuzzy logic is employed to rank the data. The scheme has a lower packet loss rate, and critical data are transmitted to the MS effectively, but the security perspective needs to be addressed [60]. To resolve the hotspot problem in WBANs, a temperature-aware scheme was presented, which considers the temperature of nodes before selecting the CH. The clustering is performed for data aggregation based on fuzzy logic. The fuzzy logic considers route breakage, residual energy, and nodes having the same neighbors. Similar nodes lie in the same cluster and aggregate data efficiently. The proposed scheme consumes low power while transmitting data to the coordinator, but complexity increases in high-level heterogeneous environments [61]. To understand the health situation of patients, the K-edge mechanism was introduced in a heterogeneous health-sector environment. In the initial phase, fuzzy logic is used to know the heart condition of patients. Then, a CNN mechanism is employed to determine the respiratory condition of patients. By using multianalysis and Mamdani fuzzy output, the patient’s condition is evaluated. The K-edge implementation provides better results regarding patient condition. The results demonstrated an accuracy of 98.68% for respiratory conditions and was suitable for performance in resource-restricted environments [62]. In another method, to increase the efficacy of data transmission in the WBAN context, the aggregation node dynamically selects the aggregation frame. The data are classified into seven user priorities (UPs): UP7 has the highest priority, and UP0 has the lowest priority. The data are transferred to queues and sent to the central hub, where they are transmitted to the main server. Queue 0 contains emergency data, which are dispatched without delay. At the same time, aggregation occurs at Queues 1 and 2. For selecting an aggregation frame, the DQN algorithm is used. The hub node offloads the tasks and sends them to the medical server [63].

### 2.2. Schemes Dealing with Data-Aggregation-Based Delay

During this aggregate process, hurdles like high energy consumption and delays occur, reducing network efficiency. An optimal partial aggregation (OPA) scheme was presented to resolve this issue. In this scheme, a node with a higher lifetime is considered an aggregator. A multiswarm fruit fly optimization algorithm (MFOA) is used to extend the aggregator node’s lifespan. Subsequently, an enhanced version of time to task (ToT) is used to reduce the delay issue in the aggregation of data. To find the quickest path to send data from source to destination, a nondominated sorting gravitational searching algorithm (NSGSA) is used. For simulation, an NS-2 simulator was used. The results showed that OPA has a longer network life span, low delay, and high throughput, and utilizes resources efficiently compared with previous techniques [64].

Game theory and an ant-colony-based data gathering technique (GTAC-DG) scheme were introduced to maintain the energy level of nodes to improve system performance for a long time. In the first level, the game theory idea is applied, in which all nodes in a network act as players, and nodes with high residual energy and load have a high probability of becoming rendezvous points (RPs), while other nodes become candidates. During the second stage, the best optimum path is selected. An enhanced version of ant colony optimization determines an appropriate trajectory for the mobile sink to reach the RPs. RPs having a high load are given significant importance to visit by the MS during data collection. The proposed scheme was simulated in MATLAB 2018b. This scheme has better network life and economic resource consumption and fewer dead nodes than former schemes. The drawback of this scheme is that implementing both algorithms increases computation overhead and makes it costly to deploy [65].

## 3. System Model and Problem Statement

In this section, the system model of the proposed scheme is discussed. The scenario of the system model is presented in Figure 1. The nodes are randomly organized in a heterogeneous manner to collect different types of healthcare data from clusters. The patients are equipped with wearable medical sensor equipment to be monitored. In level 1, these heterogeneous sensors collect data readings from the patient’s body and transmit them to the aggregating (parent) node. In level 2, multiple aggregators share their data with fog servers, and data are stored there, accessed by healthcare professionals to check the patient’s underlying health conditions. When sensors continuously transmit data, they contain duplicate data; for example, when a patient’s temperature or heartbeat does not fluctuate, the same readings are frequently shared. Boolean values are initialized to reduce this data redundancy, which transmits zero when the data are the same as earlier. In level 3, after the data duplication issue is resolved, the data are shared with a countrywide cloud server. Authenticated medical professionals or users can access the specific health information stored in the cloud servers. When an authenticated user requests detailed information, the edge nodes first receive the request. The edge nodes send the necessary information to the requested device if the required data are available. Otherwise, the edge nodes retrieve the needed data from the cloud repositories.

As in the existing scheme, the data aggregation process in a heterogeneous environment uses a tree-based approach with bottom-up scheduling to maintain aggregation freshness. Parent nodes are selected from the candidate set having a direct link to the child node based on the number of the least dynamic neighbors. In such cases, when more than one candidate node has the same number of active neighbors, fuzzification is performed; afterward, min–max normalization is used to scale-up the weights on the edges of nodes in the graph. The consequences on the edges of the nodes are normalized using min–max normalization. These weights work as membership values, and the node with the lowest membership value and having a direct link to the child node is chosen as the parent node. But in aggregation, it shares the same readings of parameters again and again. For example, it repeatedly shares the same temperature value in an entire day, even if it remains the same or within the normal range.

Sharing duplicate data occasionally results in a negative impact on WSNs. Frequently sharing the same data wastes network resources like bandwidth and battery power. It enhances the communication overhead and the additional energy consumption as well. Repeatedly transmitting the same data among many sensor nodes in a network can cause congestion, leading to decreased reliability and performance. It also decreases network lifetime, reduces data quality, and causes late recognition of anomalies. To overcome these drawback, data size needs to be reduced, which needs to be accommodated in existing techniques [40].

## 4. Proposed Solution

The proposed scheme, the fuzzy data aggregation system (FDAS), aims to reduce energy consumption while transmitting data. This study considered the data generated by sensors in healthcare environments. Patients wear different healthcare devices so that their blood pressure, pulse rate, and sugar levels can be monitored and immediate action can be taken in the event of an emergency. All these wearable sensors are different, measuring data in various formats, so the network is heterogeneous, where nodes generate multiple data. A synchronized tree-based mechanism is proposed for better aggregation of these heterogeneous data. The bottom-up process of data aggregation from nodes is achieved, and selecting a suitable parent is mandatory for the economic consumption of nodes’ energy.

To achieve these objectives, two main phases are the control and the data phases. During the control phase, nodes select a slot and an appropriate parent. In the data phase, data are sent through the tree to the parent node created in the control phase. In a network, nodes receive data packets and send them, too. The first scenario deals with nodes receiving data packets; it has complete information about (i) the total number of arriving data packets and (ii) the type of data packet. In addition to this, the node knows about the attributes of the data packet created independently. When nodes start selecting their time slot to transmit data to the parent, they need to perform some tasks, including (i) knowing the type and number of data packets, (ii) labeling the nodes, and (iii) giving weights to edges. For weight, the distance between nodes is calculated. In the second case, when the node transmits a data packet, the transmitter node knows (i) the total number of departing packets and (ii) the attributes of the data packets. The node performs a suitable parent selection for every outgoing data packet, so there is a probability that a different parent is selected for every outgoing data packet.

### 4.1. Appropriate Parent Selection

In a heterogeneous network, selecting an appropriate parent node is challenging because there is a high chance that the children of a node generate different types of data packets. If the parent aggregation node is selected efficiently, energy consumption is reduced. In a node, energy is utilized when control data packets are swapped while selecting the slot and parent. The efficient selection of the parent node effectively aggregates data from nodes, which directly impacts total slot usage, control messages, and energy utilization in both the control and data phases. In this stage, the energy utilization of node i $\left(E_{i}^{c}\right)$ and $E^{C}$ is determined using Equations (Equation 5) and (Equation 6) [66]. $E^{\mathrm{initial}}$ represents the starting energy level of the nodes in the network; $E_{i}^{residual}$ denotes the residual energy at the end of this phase. Equation (Equation 6) determines the average energy consumption in the control phase $\left(E^{C}\right)$ of the nodes:(5)$$\begin{array}{cc} \; & E_{i}^{c}=E^{\mathrm{initial}\;}-E_{i}^{\mathrm{residual}\;}\;i=1,2,3,\ldots ,n \end{array}$$
(6)$$\begin{array}{c} E^{c}=\frac{\Sigma_{-1}^{n}E_{i}^{c}}{n}\;i=1,2,3,\ldots ,n \end{array}$$

The flow of different algorithms is shown in the block diagram in Figure 2. The set of notations is presented in Table 1.

The parent node selection is not a solo mechanism: slot and parent selection are performed together to reduce energy consumption and enhance network lifetime. To select an appropriate parent node, first of all, nodes having direct links to the child node are checked. These sets of nodes are named as candidate nodes. Thier total dynamic number of neighbors is counted for all the candidate nodes. The node with the least-active neighbors is considered the parent node. In the scenario where two candidate nodes have the same number of dynamic nodes, the normalized weights for edges are calculated using Equation (Equation 7) [67], where *v* is the value to be scaled, Min(A) is the minimum original value, Mox(A) shows the maximum original value, now M(w)A) is the new minimum value in the new range of data, ngwcos(A) is the new minimum value in the new range of data, and vl shows the scaled value of *v*.
(7)$$v1=\frac{v-\mathrm{Min}(A)}{\mathrm{Max}(A)-\mathrm{Min}(A)}\left(\mathrm{newMax}\left(A\right)-\mathrm{new}\mathrm{Min}\left(A\right)\right)+\mathrm{new}\mathrm{Min}\left(A\right)$$

In the fuzzifier, the crisp input values are given to the FIS to obtain an accurate, crisp output after defuzzification. The linguistics used for input values are low, medium, and high. For the fuzzy inference system, three input values are considered based on these input values; one output is generated. The input set consists of residual energy, relative node connectivity, and load on the node. The input value is explained using the AND operator preceding with an IF statement. In the context of the FDAS, the triangular member function is employed because of its simplicity, ease of interpretation, and effective computation. The triangular member function is appropriate for observing gradual variations in the selected parameters. These parameters are residual energy, node connectivity, and load on nodes, which are crucial factors in the context of the healthcare sector. By applying triangular membership functions, the system remains obvious, interpretable, and well suited to the domain’s necessities without excessive complexity. The Mamdani fuzzy system (type 1) is used in the proposed scheme. It can handle ambiguity and vagueness in decision making from various potential outcomes. In the proposed solution, where node types are wide in range, generating data readings in different formats, the Mamdani fuzzy system checks the linguistics of the input parameters. Finally, it assists by selecting an appropriate parent node. The 27 rules used as FISs are explained in Algorithm 2. Residual energy is the remaining energy in the sensor node after performing some operation like sensing or transmitting bits of data to the parent node. The member function of residual energy with its linguistics is shown in Figure 3.

The second input is a load on the candidate node, which is an important metric as it shows volumes of data on a node to be processed as communication overhead while sharing data with other nodes. In Figure 4, the member functions for this input are shown. The third input is relative node connectivity (RNC); this metric defines how well a node is connected to its subgraph and the nodes at a higher level. The functions for this input are shown in Figure 5.

Figure 6 elucidates the output obtained according to input to the fuzzy system. It contains nine triangular MFs: very low, low, quite low, lower medium, medium, higher medium, high and very high.

In the defuzzifier, the final crisp output is calculated by the center of the area (CoA) shown in Equation (Equation 8) [67]. A node having high residual energy, higher RNC, the lowest load, and the minimum weight has the highest chance of being selected as a parent node. $\sum_{i=1}^{rus}\mu \left(k\right)$ is the sum of the membership degree of each node, and u(μ(k)) is a numerical value of obtained at level *k*, which denotes the certain node. The detailed procedure of the parent selection node is shown in Algorithm 1.
(8)$$\mathrm{CoA}=\frac{\sum_{i=1}^{\mathrm{rules}\;}\mu \left(k\right)\cdot u\left(\mu \left(k\right)\right)}{\sum_{i=1}^{rules}\mu \left(k\right)}$$

**Algorithm 1** Parent Selection at Control Phase
1:**Input:** total number of sensor nodes2:**Output:** Parent node3:**if** root=Nul **then**4:    Generate root5:    **End**6:
**end if**
7:**while** true **do** adding edges8:    **// s denotes source, d denotes destination, and w denotes weight**9:    function addEdge(int s, int d, int w)10:
**end while**
11:// during control phase12:**for** each node **do**13:    **if** t_slot_n[i] == a_slot **then**14:        Candidate_node[] = direct_link(C_node[]))15:        set n=**function** choose_aggregator(Candidate_node[])16:        // For checking dynamic neighbors17:        Neigh[] = count_dynamic(Candidate_node[]))18:        min_neigh=node[0].neigh[0]19:        **for** i=1; i<=size; i++ **do**20:           **if** (node[i].neigh[i] < min_neigh) **then**21:               parent= node[i].neigh[i]22:           **else**23:               parent= min_neigh24:           **end if**25:        **end for**26:        return n27:        **End function**28:    **end if**29:
**end for**



In step 3–16, In steps 3–16, firstly, check that the tree-based WSN has a root tree; if it is empty, a root node is created. The add edge function is used to create edges between nodes, and weight is determined by calculating the difference between two nodes. The nodes are assigned a unique time slot to send data to avoid collision. A node in its allotted time slot checks the set of nodes directly linked to child nodes. These nodes are termed as candidate nodes. For these candidate nodes, the dynamic neighbors of each node are checked. The value of the first node is saved in the variable min_neigh. In steps 17–23, for loop starts that iterate over all the nodes, for all the nodes, it is checked that the value in the node is lower than that of the min_neigh. If the condition is true, then that node becomes the parent node; otherwise, min_neigh node remains the parent node. Finally, the function returns the value of the variable n, which was assigned in line 12. Afterwards, the function ends there.

In case two candidates have the same number of dynamic neighbors, fuzzy logic is employed for the selection of the parent node from the candidate set. For selecting the parent node, three fuzzy-based inputs are considered including residual energy (RE), relative node connectivity (RNC), and load on node (LN). The probability of nodes being elected as the parent node from the candidate set is determined using Algorithm 2. The membership values of the nodes are determined using the fuzzy inference system, represented as FIS (node[].RE, node[].RNC, node[].LN). $\sum_{i=1}^{rules}\mu (k)$ denotes the summation of the membership degree of each node, while u(μ(k)) is a numerical value obtained at round k. Through $CoA=\frac{\sum_{i=1}^{rules}u(k)\cdot u(\mu (k))}{\sum_{i=1}^{rules}\mu (k)}$, defuzzification is performed, and a crisp output is obtained. To scale-up weights in the range of [0, 1], min–max normalization is performed. The node having the best FIS result and the lowest weight is selected as the parent node. Algorithm 2 takes residual energy, relative node connectivity, and load on candidate node of the candidate nodes as the input parameters and, based on node status, a linguistic output is generated that shows the chances of selection of a parent node from the candidate set. The FIS designed for the proposed scheme is shown in Figure 7.
**Algorithm 2** Fuzzy algorithm and rules for FIS.1:**Function** FIS (RE, RNC, LN, PSCN) //RE: residual energy, RNC: relative node connectivity, LN: load on candidate node, PSCN: probability of selecting candidate node, CN: candidate node2:Rule 1: IF, CN (RE is high), AND (RNC is high), AND (LN is High) THEN (PSCN is higher medium).3:Rule 2: IF, CN (RE is high), AND (RNC is high), AND (LN is medium) THEN (PSCN is high).4:Rule 3: IF, CN (RE is high), AND (RNC is high), AND (LN is low) THEN (PSCN is very high).5:Rule 4: IF, CN (RE is high), AND (RNC is medium), AND (LN is high) THEN (PSCN is medium).6:Rule 5: IF, CN (RE is high), AND (RNC is medium), AND (LN is medium) THEN (PSCN is medium).7:Rule 6: IF, CN (RE is high), AND (RNC is medium), AND (LN is low) THEN (PSCN is higher medium).8:Rule 7: IF, CN (RE is high), AND (RNC is low), AND (LN is high) THEN (PSCN is medium).9:Rule 8: IF, CN (RE is high), AND (RNC is low), AND (LN is medium) THEN (PSCN is medium).10:Rule 9: IF, CN (RE is high), AND (RNC is low), AND (LN is low) THEN (PSCN is medium).11:Rule 10: IF CN (RE is medium), AND (RNC is high), AND (LN is high) THEN (PSCN as parent is low).12:Rule 11: IF CN (RE is medium), AND (RNC is high), AND (LN is medium) THEN (PSCN as parent is low).13:Rule 12: IF CN (RE is medium), AND (RNC is high), AND (LN is low) THEN (PSCN as parent is medium).14:Rule 13: IF CN (RE is medium), AND (RNC is medium), AND (LN is high) THEN (PSCN as parent is lower medium).15:Rule 14: IF CN (RE is medium), AND (RNC is medium), AND (LN is medium) THEN (PSCN as parent is medium).16:Rule 15: IF, CN (RE is medium), AND (RNC is medium), AND (LN is low) THEN (PSCN as parent is medium).17:Rule 16: IF, CN (RE is medium), AND (RNC is low), AND (LN is high) THEN (PSCN as parent is lower medium).18:Rule 17: IF, CN (RE is medium), AND (RNC is low), AND (LN is medium) THEN (probability of, CN as parent is medium).19:Rule 18: IF, CN (RE is medium), AND (RNC is low), AND (LN is low) THEN (PSCN as parent is medium).20:Rule 19: IF, CN (RE is low), AND (RNC is high), AND (LN is high) THEN (PSCN as parent is low).21:Rule 20: IF, CN (RE is low), AND (RNC is high), AND (LN is medium) THEN (PSCN as parent is Quite_low).22:Rule 21: IF CN (RE is low), AND (RNC is high), AND (LN is low) THEN (PSCN as parent is Quite_low).23:Rule 22: IF CN (RE is low), AND (RNC is medium), AND (LN is high) THEN (PSCN as parent is very low).24:Rule 23: IF, CN (RE is low), AND (RNC is medium), AND (LN is medium) THEN (PSCN as parent is low).25:Rule 24: IF, CN (RE is low), AND (RNC is medium), AND (LN is low) THEN (PSCN as parent is Quite_low).26:Rule 25: IF, CN (RE is low), AND (RNC is low), AND (LN is high) THEN (PSCN as parent is very low).27:Rule 26: IF, CN (RE is low), AND (RNC is low), AND (LN is medium) THEN (PSCN as parent is low).28:Rule 27: IF, CN (RE is low), AND (RNC is low), AND (LN is low) THEN (PSCN as parent is very low).29:return result30:**End function**

### 4.2. Data Packet Transmission in Data Phase

In the data phase, data are sent to the aggregator and from the aggregator to the fog server. These data contain duplicate information, which increases energy consumption during transmission. By using Equations (7) and (8), the energy utilization is calculated during the data phase $\left(E^{D}\right)$. To reduce the energy consumption in the proposed system, duplicate data are checked before saving to local devices and shifted to a cloud server. As in an existing scheme, FAJIT, when the entire network is homogenous (when all nodes are of similar type), perfect data aggregation happens. The increase in attributes may decrease the aggregation factor. The attributes may include temperature, humidity, solar radiation, and humidity in cases where heterogeneity is high. Finding an appropriate parent node where suitable data collected from the nodes can be aggregated is problematic. In addition to this, transmitting the same data repeatedly increases communication overhead and lowers the method’s performance. The proposed solution considers the values of nodes that are communicating in the same or in a similar range; for such values, a Boolean value of zero or one is transmitted, where zero represents that value is the same as that shared previously; there is no variation in data, so there is no need to write the entire value of 16 or 32 bits. Instead, 1 bit is sent to save the number of bits transmitted in the networks. For example, if a patient’s pulse rate is the same, and there is no variation in its reading, it is better, to avoid transferring duplicate data frequently, to share zero digits to show the data are the same as before. So, as heterogeneity increases, attributes and variation increase, but the data size reduces. This helps with lessening energy consumption and aggregation factors and reduces the message size.

Along with this, it significantly reduces the effect of an increase in the number of attributes on the performance metrics, including aggregation score, transmission cost, and energy consumption. In Algorithm Algorithm 3, the proposed mechanism to reduce energy consumption is presented.

From steps 3–12, all nodes send their data to aggregating nodes, which transmit them to the nearest fog servers, where data are processed and checked for duplicates. If duplicate data are present, they are converted to Boolean format zero to show no variation, and then saved to a local repository so that doctors or healthcare professionals can access them. Afterward, when data are free from duplicates, they are sent to a cloud server and stored there. Table 2 shows the optimized fuzzy-centered table for electing a parent node from a set of candidates.
**Algorithm 3** Transmission of Packets At Data Phase1:**Input:** Set of ANs2:**Output:** Redundancy-free data storage to cloud3:**For** each node4:transmit_data(n[i]) to ANs5:**End**6:transmit_info(ANs) to fog server7:**If** transmit_info_new == transmit_info previous **then**8:Send Boolean digit 0 for duplicate data9:Save to local storage device and transmit it to cloud server10:**else**11:Save to local storage device and transmit it to cloud server12:**End**

## 5. Results and Analysis

To evaluate the performance of the proposed scheme, FDAS, extensive simulations were performed using NS-2.35. The nodes were placed randomly on the premises of the network. The whole area was divided into grids of 20 × 20. Between each two horizontal and vertical grid points, constant distance of 156 m was considered. Medical sensors were deployed on each patient as per the position of that sensor on the body. The patients inside the wards were deployed on a grid, whereas the outside patients were deployed per a Gaussian distribution. Arranging medical sensors on patients in a grid pattern inside the ward and using a Gaussian distribution for patients outside allowed for the efficient monitoring and management of patients’ health conditions. The mobility patterns of the patients were also considered while designing the sensor arrangement mechanism to efficiently accommodate various healthcare scenarios. The data transmission range for all types of sensors was kept to 30 m, while nodes produced a data packet after each 10 s. The time allotted for simulation was 2500 s to examine network performance for a prolonged period. A total of 300 nodes were placed in a network region of 3000 × 3000 m to observe different node attributes in a heterogeneous environment. For sending and receiving data packets, 0.5819 µj and 0.049 µj were considered for utilizing energy economically in the transmission range of 30 m. The values of the simulation parameters were adjusted considering real-world scenarios. The simulation time assigned to evaluate proposed scheme is 2000 s. For each scenario, simulation is performed various times and average were calculated. TCL files contained the node configurations and their arrangements, while for receiving and transmitting data packets, a separate class was created using c language. To obtain results from trace files, AWK script files were used.

The value assumed for probability was 0.5, which indicated a 50% chance that a node existed in the grid. As a heterogeneous environment was considered for the proposed scheme, four scenarios were used to evaluate the performance of the FDAS. When the number of attributes was one, it indicated that the network was homogenous, as all sensors were of the same type. The attributes showed the kind of nodes generating data per environment. In the first scenario, when the number of attributes was two, it indicated two types of nodes in the environment generating data. In the second scenario, four types of sensors were present in the network. As the attributes started increasing, the network became highly heterogeneous. All the attributes assigned to the nodes had equal probability. As in the third scenario, when the number of attributes was eight, such as $A_{1},\;A_{2},\;A_{3},\;A_{4},\;A_{5},\;A_{6},\;A_{7}$, and $A_{8}$. Then, $\rho^{Aj}=\frac{1}{8}=0.125$ shows that any attribute allotted to a node had an estimated probability of 0.125. For validating the effectiveness of the proposed scheme, crucial metrics were considered, and the results were compared with those of some existing schemes: FAJIT, DICA, and DICA _EXTENSION.

Table 3 shows the simulation parameters and values.

### 5.1. Effect on Average Aggregation by Number of Attributes

This metric represents total number of data packets accumulated at the parent node before being transferred to the fog server. When there is only one attribute in the network, all schemes performed well. But as the heterogeneity started to increase, the network became complex, and average aggregation started falling. When there were two attributes, the effectiveness DICA and DICA _EXTENSION started decreasing; the performance of FAJIT also decreased as the network complexity increased. FDAS achieved better aggregation even when the network started becoming more complex. The reason for this better aggregation achieved by the proposed scheme is that the attributes of every packet are considered before selecting the appropriate parent node. This increases the probability that data packets of type t are aggregated at the selected parent node; ultimately, parent nodes have to transmit a low number of packages farther, so when a suitable parent is selected, the aggregation factor remains stable even when heterogeneity starts increasing. In addition, in the case of equal dynamic neighbors, parameters are set intelligently to choose a parent node, which is not performed in existing schemes. In DQN-FATOA, the aggregation factor is better due to the scheme’s intelligent optimization approach. The aggregation node dynamically adjusts the frame length and the number of tasks offloaded to attain better aggregation.

For determining average aggregation, the summation of each node’s aggregation was considered. Figure 8a shows the average aggregation of all schemes for different numbers of attributes. On the *X*-axis, the number of attributes is shown, which is basically the types of nodes in the environment, while the *Y*-axis expresses the aggregation level in percentage. When the number of attributes is four, four different types of nodes are present in the WSN environment. The FAJI achieved an aggregation of 0.52, DQN-FATOA attained 0.62, DICA attained 0.35, DICA _EXTENSION obtained 0.39, and proposed scheme FDAS obtained highest aggregation level of 0.87. FDAS maintained a stable level of 0.24%,0.30%, and 0.27% compared with FAJIT, DICA and DICA _EXTENSION when the network was more complex (up to 16 attributes).

### 5.2. Energy Utilization during Control Phase

The control phase refers to the period when the network is created, nodes are set, and slots are selected, as well as suitable parents are selected to aggregate the data. It is the most important phase because the network performance of the data phase is mainly based on the control phase. If the control overhead increases during the selection of slots and parents, the energy utilization is also high. As energy depletion during control overhead has a direct impact on control phase, the appropriate parent node selection reduces the consumption of energy because it leads to choosing fewer slots for transmission. In DQN-FATOA, deep Q learning decisions are made for energy-intensive activities, for instance, parent and slot selection. Therefore data consumption is reduced in this phase. As the proposed scheme, FDSA, intelligently selects the parent node at each level by knowing the type of node, the energy consumption is lower than that of FAJIT, DICA, and DICA _EXTENSION. In Figure 8b, the *X*-axis presents the number of attributions, and the *Y*-axis presents the energy consumption in micro joules (µJ). For two attributes energy consumption is 88, 60, 108, 103, and 50 in FAJIT, DQN-FATOA, DICA, DICA _EXTENSION, and FDAS, respectively. FDAS reduced the energy utilization by up to 38%, 58%, and 53% compared with FAJIT, DICA, and DICA _EXTENSION.

### 5.3. Energy Utilization during Data Phase

Once slot and parent selection is completed, data are transmitted during the data phase. In a WSN environment, sensors generate the same data when there is no variation in the environment, and this duplicate data transmission increases energy consumption. The proposed scheme, FDAS, compresses the duplicate data by converting them into Boolean digits and transmitting only one bit, zero, to indicate redundant data. As the number of attributes increases, energy usage rises. But, the data conversion feature of FDAS provides significant improvements in energy utilization. So, the increase in the number of attributes does not have much impact on the energy factor due to the replacement of redundant data in the Boolean system, which eventually lessens the energy usage compared with that of previous schemes. In DQN-FATOA, the distribution of computational tasks is balanced by the DQN mechanism, so nodes are not overburdened, which reduces energy consumption. No such mechanism is used in the other schemes. FAJIT consumes more energy in this phase because of finding a suitable parent node. DICA and DICA_EXTENSION select the nearest nodes as parents and consume less energy. The performance of FDAS and existing techniques is shown in Figure 9a. The *X*-axis shows the number of attributes; the *Y*-axis presents the energy consumption during the data phase in (µJ). For 12 attributes, which produces a higher level of heterogeneity, the energy consumption is 34 for the proposed scheme µJ, 60 for FAJIT µJ, 45 for DQN-FATOA µJ, 72 for DICA µJ, and 67 for DICA µJ.

### 5.4. Effect of Number of Attributes on Schedule Length (SH)

Schedule length (SH) is defined as the selection of the exclusive slots needed to schedule data from a set of sensor nodes to the aggregator node. The schedule length is inversely proportional to the aggregation factor. If aggregation is low, schedule length (SH) is long. Similarly, if aggregation is higher, then the schedule length is shorter. The schedule length also varies in a heterogeneous network. In a subtree where eight types of nodes are generating data, finding a suitable parent is more difficult than in a subtree where only two types of nodes are present. The reason for the shorter schedule length is the dynamic capability of DQN-FATOA to aggregate frames and make offloading decisions. The length of the aggregated frames are observed and combined with a smaller one. This feature permits proficient resource utilization and reduces schedule length. The proposed Figure 9b scheme produces better aggregation that minimizes the total data packets passing through a TDMA cycle. The *X*-axis shows the number of attributes; the *Y*-axis presents the average SH of all schemes in bytes. When the number of attributes was 16, FDAS attained an average value of 235, DQN-FATOA obtained an average value of 244, FAJIT obtained an average value of 250, DICA obtained an average schedule length of 275, and that of DICAEXTENSION was 285. FDAS provides improvements of 30%, 10%, 22%, and 5% over FAJIT, DICA, DICA, and DQN-FATOA, respectively.

### 5.5. Required Number of Transmission Slots

When increased data aggregation occurs, fewer data packets are generated, and fewer transmission slots are required to transmit data. The appropriate data aggregation factor, merging smaller frames with other frames, permits more data to be combined and sent, which simultaneously results in fewer transmissions. The proposed scheme, FDAS, improves the allocation of transmission slots via intelligent parent node selection, efficient data aggregation, and energy preservation by reducing the data packet size. This results in a reduced number of transmission slots required to transmit data. FDAS attained a lower number of transmission slots for all attribute cases. In Figure 10a, the values on the *X*-axis represent the number of attributes, and the *Y*-axis represents the number of transmission slots. For 16 attributes, 930, 980, 1000, 1200, and 1100 transmission slots were needed to forward data in FDAS, DQN-FATOA, FAJIT, DICA, and DICA _EXTENSION.

### 5.6. Effect of Number of Attributes on Control Overhead

The control overhead of node i refers to the total control messages transmitted by that node. In the control phase, when slot and parent selections are performed, control messages are exchanged with neighbors so that a collusion-free procedure is carried out. The control messages consist of a chain of request, reply, schedule, and forbidden. For selecting more slots, more chains of these control messages are exchanged. As control overhead increases, energy consumption also increases. Therefore, to reduce control overhead and minimize the energy consumption factor, slot and parent selections are performed simultaneously during the control phase of FDAS. Fewer selected slots leads to the lesser generation of control messages. The *X*-axis shows the number of attributes, and the *Y*-axis presents the control overhead in bytes/second. Figure 10b shows that when the number of attributes increases, control overhead also increases, which shows that attributes have a direct impact on control overhead. For 10 attributes, the control overhead lies in the range of 1000, 10,060, 10,090, 9050, and 9000 for FAJIT, DICA, DICA_EXTENSION, DQN-FATOA, and FDAS.

### 5.7. Discussion

FDAS was compared with FAJIT, DQN-FATOA, DICA, and DICA_EXTENSION for a complete analysis. The average aggregation is better with FDAS as it selects a parent node with the same attribute, which takes less time to aggregate and transmit. DQN-FATOA adjusts frame size for data aggregation and aggregates small data packets to use resources effectively. FAJIT performs well when the number of attributes is small, but when the variation in attributes is high, FAJIT’s performance starts dropping and reaches the same level as that of DICA and DICA_EXTENSION. Fuzzy logic is employed when the number of dynamic neighbors is the same, and the most crucial factors are selected to determine a suitable parent node, which reduces energy utilization in the control phase in the proposed scheme. However, in DICA and DICA_EXTENSION, no fuzzification is performed. Though a fuzzy system is considered in FAJIT, no suitable input functions are described to overcome this issue. Slots and parents are selected simultaneously in FDAS to lessen the control overhead. The control overhead is low in DQN-FATOA due to task offloading as required in real-time scenarios, which overcomes the necessity of transmitting control signals frequently. As a result, the number of control messages between nodes and the hub is minimized, which reduces the overhead. As a whole, the network can effectively transmit and process data without unnecessary control messages. FAJIT also performs better than the other two schemes in selecting fewer slots because of its better aggregation. During the data phase, in DQN-FATOA, no node is overburdened and data are effectively handled, so energy consumption remains economical. In FAJIT, energy consumption is high when the number of attributes increasesbecause selecting a parent of the same type becomes difficult. In contrast, DICA and DICA_EXTENSION select the nearest node as the parent node. The parent node chosen is at the same or one level distant from the sensor node. This results in lower energy consumption in the data phase. In FDAS, the aggregated data contain duplicate data that fall in the normal range; they are not transmitted in their full form of 32 or 16 bits. Instead, a Boolean zero digit is transmitted, considerably reducing energy consumption; even if the types of nodes increase or the environment becomes more complex, FDAS maintains better performance. Overall, FDAS and DQN-FATOA perform well on most of the metrics. FAJIT maintains good performance, but its energy consumption is slightly higher during the data phase. At the same time, DICA and DICA_EXTENSION have a longer schedule length, are highly affected by environment complexity, and are unsuitable for use in resource-constrained scenarios.

## 6. Conclusions

In this paper, a scheme, FDAS, that removes duplicate values in the data generated by sensors in a heterogeneous environment was presented. Duplicate data increase the size of the data packet; reduce the aggregation factor, ultimately increasing the transmission slots required for sending data; and increase control overhead, reducing network lifetime and degrading the overall performance of the WSN. By considering this issue, we developed FDAS to convert duplicate data with a Boolean digit of zero, significantly improving heterogeneous networks. In addition, better parent node selection is performed by including the most crucial parameters in the FIS system, which enhances the efficiency of FDAS. The scheme was simulated using NS 2.35, and the results showed that the proposed method has better performance, as its energy utilization is lower in both phases, overhead is low in all attribute scenarios, transmission slots usage is decreased, and SH is reduced compared with those of FAJIT, DQN-FATOA, DICA, and DICA_EXTENSION. In the near future, the first node death (FND), half node death (HND), and last node death (LND) of the proposed scheme will be examined, along with fuzzy logic and other algorithms, such as the dragonfly algorithm (DA), to aggregate data from trees to further enhance the durability of heterogeneous WSNs. 

## Figures and Tables

**Figure 1 sensors-23-07799-f001:**
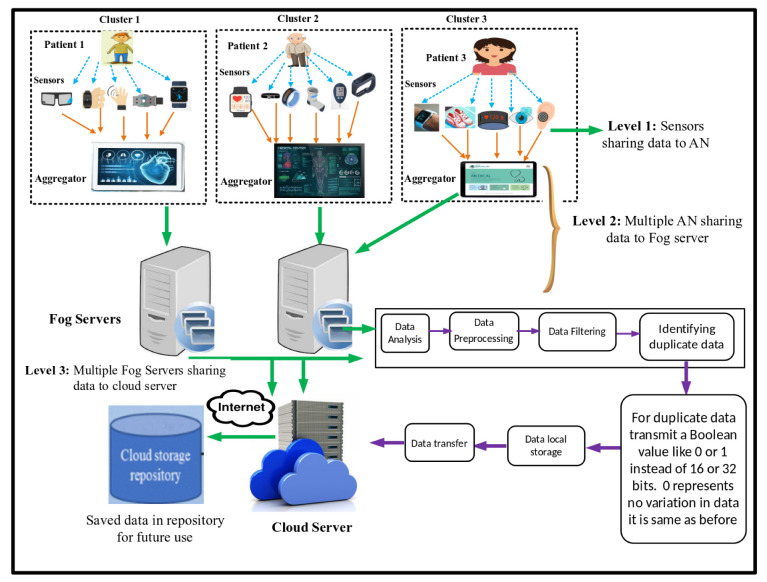
System model of proposed scheme.

**Figure 2 sensors-23-07799-f002:**
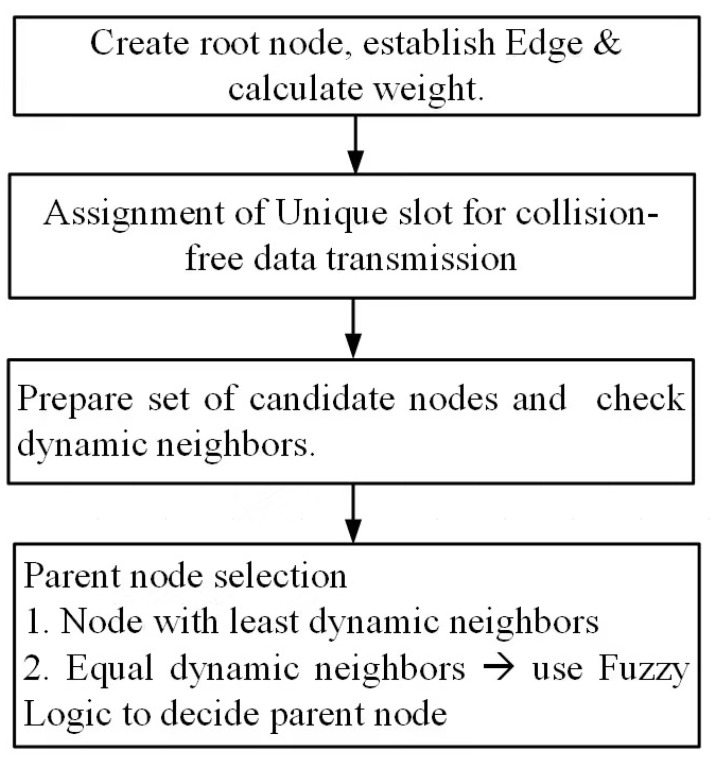
The flow of different algorithms.

**Figure 3 sensors-23-07799-f003:**
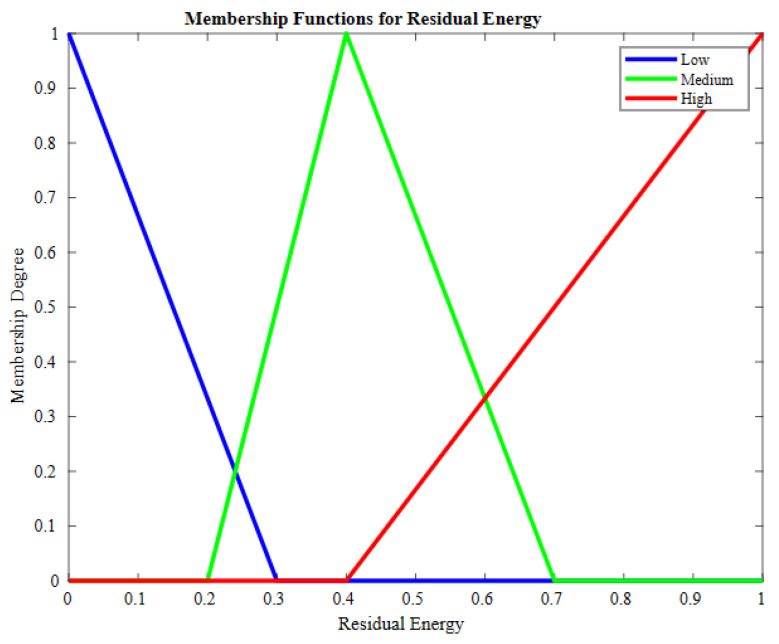
Member function for residual energy.

**Figure 4 sensors-23-07799-f004:**
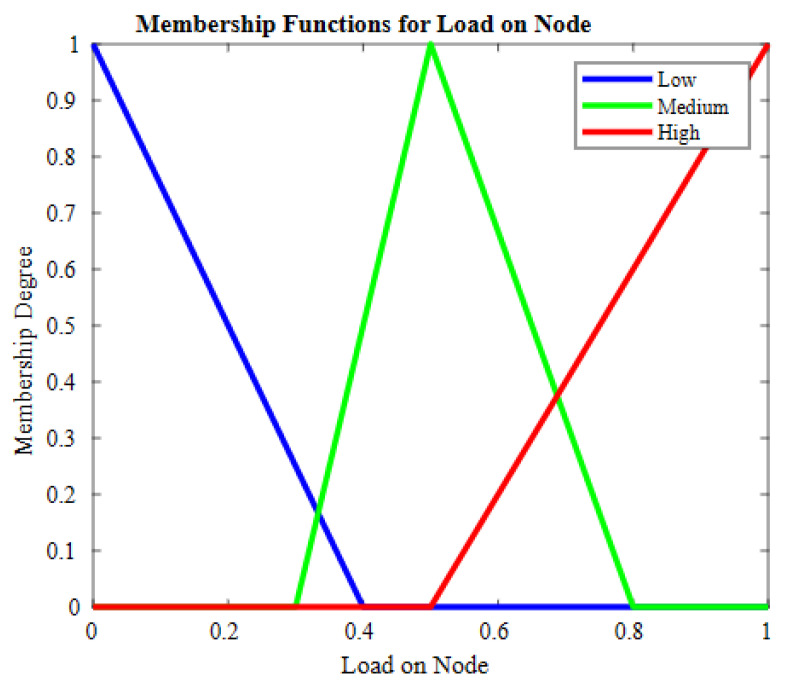
Member function for load on node.

**Figure 5 sensors-23-07799-f005:**
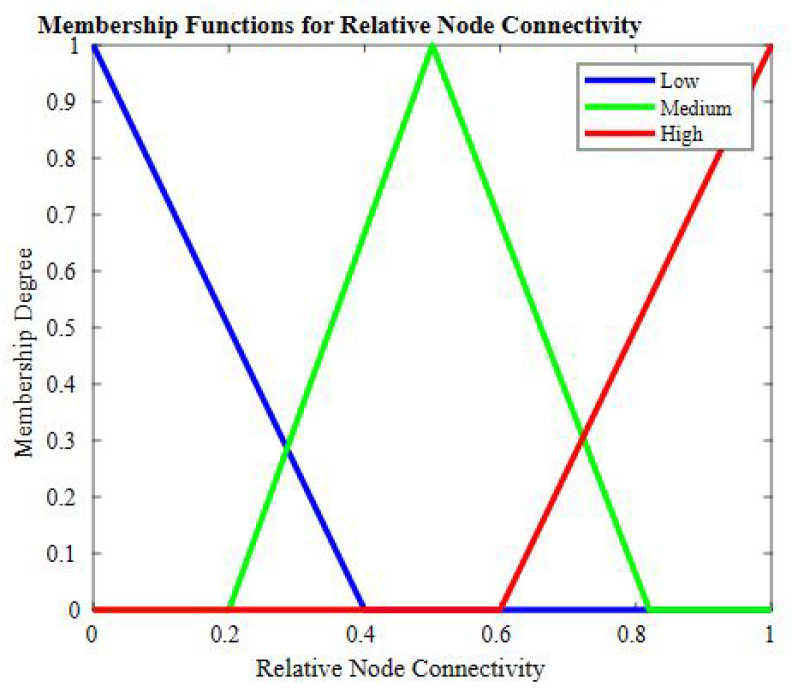
Member function for relative node connectivity.

**Figure 6 sensors-23-07799-f006:**
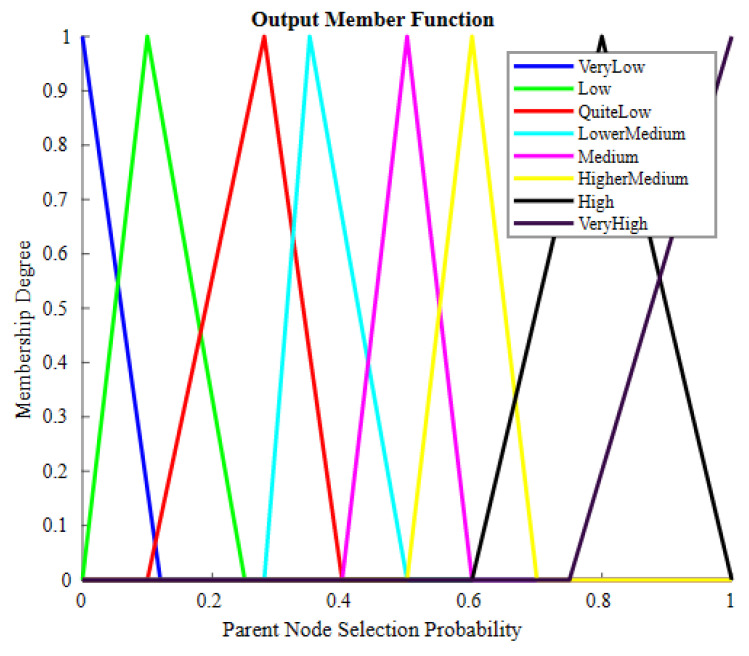
Output member functions.

**Figure 7 sensors-23-07799-f007:**
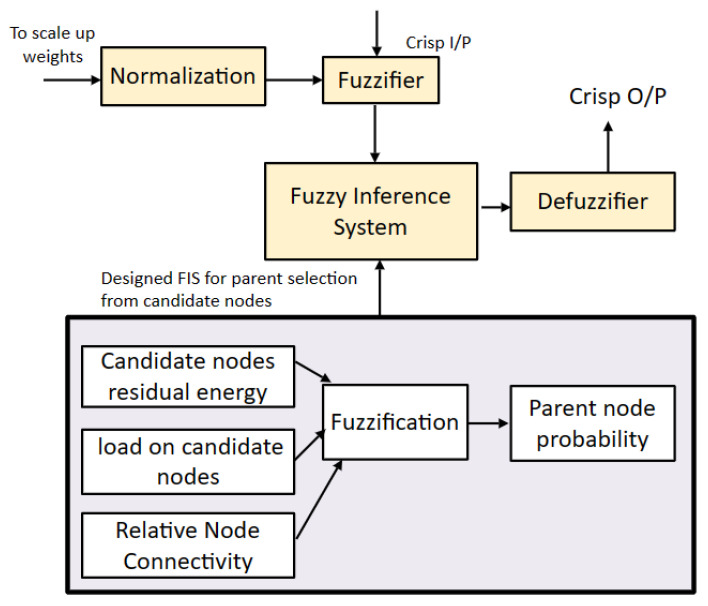
Designed FIS for proposed scheme.

**Figure 8 sensors-23-07799-f008:**
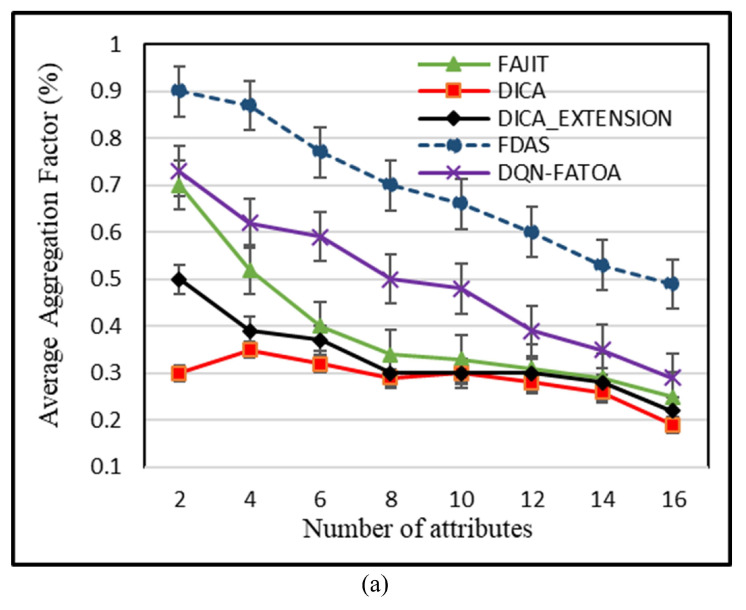
Effect on average aggregation (**a**) and energy utilization during control phase (**b**).

**Figure 9 sensors-23-07799-f009:**
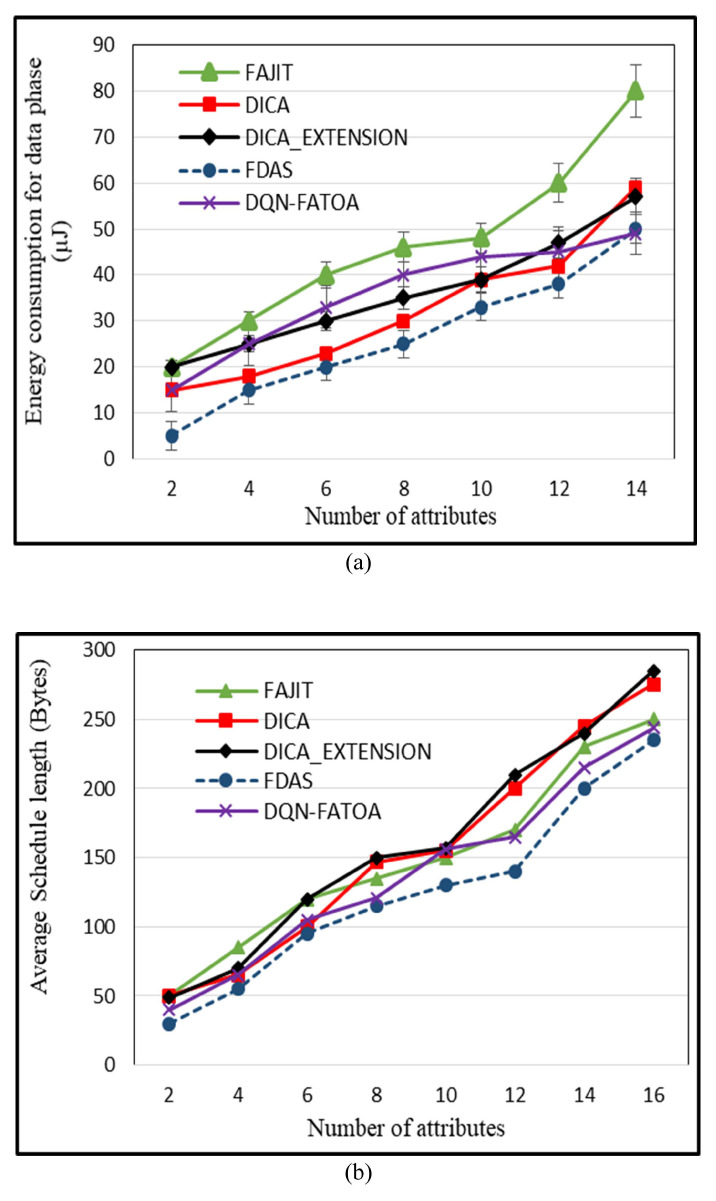
Energy utilization in the data phase (**a**) and effect of attributes on schedule length (SH) (**b**).

**Figure 10 sensors-23-07799-f010:**
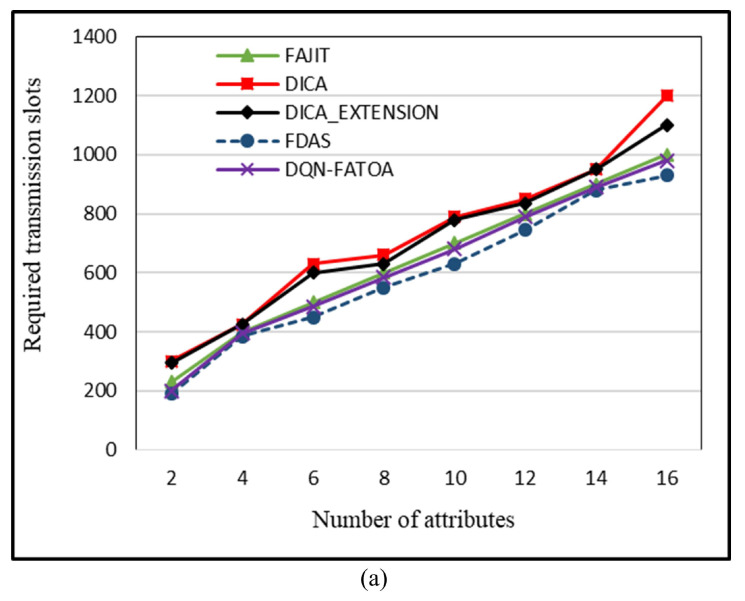
Required number of transmission slots (**a**) and effect of number of attributes on control overhead (**b**).

**Table 1 sensors-23-07799-t001:** List of notations.

Sr.	Notation	Description
1.	t_slot_n[i]	Time slot assigned to node i
2.	C_node[]	Set of child nodes
3.	ANs	Aggregating nodes
4.	FIS	Fuzzy inference system
5.	a_slot	Assigned time slot
6.	neigh[i]	Neighbor node i
7.	FL	Fuzzy logic

**Table 2 sensors-23-07799-t002:** Optimized Fuzzy-centered table for electing parent node from set of candidates.

Sr.	Residual Energy	RNC	Load on Node	Crisp Output
1	High	High	High	High Medium
2	High	High	Medium	High
3	High	High	Low	Very High
4	High	Medium	High	Medium
5	High	Medium	Medium	Medium
6	High	Medium	Low	Higher Medium
7	High	Low	High	Lower Medium
8	High	Low	Medium	Medium
9	High	Low	Low	Medium
10	Medium	High	High	Low
11	Medium	High	Medium	Low
12	Medium	High	Low	Medium
13	Medium	Medium	High	Lower Medium
14	Medium	Medium	Medium	Medium
15	Medium	Medium	Low	Medium
16	Medium	Low	High	Lower Medium
17	Medium	Low	Medium	Medium
18	Medium	Low	Low	Medium
19	Low	High	High	Low
20	Low	High	Medium	Quite${}_{Low}$
21	Low	High	Low	Quite${}_{Low}$
22	Low	Medium	High	Very Low
23	Low	Medium	Medium	Low
24	Low	Medium	Low	Quite Low
25	Low	Low	High	Very Low
26	Low	Low	Medium	Low
27	Low	Low	Low	Very Low

**Table 3 sensors-23-07799-t003:** Simulation parameters and values.

Parameters	Values
Simulation Time	2500 s
No. of deployed nodes	300
Arrangement of nodes	Random
Transmission range of data packets	30 m
Dissemination power utilization	0.5819 µj
Received power utilization	0.049 µj
Size of network	3000 m × 3000 m

## Data Availability

Not applicable.

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
