# Peer review of "Fuzzy-Based Efficient Healthcare Data Collection and Analysis Mechanism Using Edge Nodes in the IoMT"

_sensors, 2023, doi:10.3390/s23187799_

Round 1

Reviewer 1 Report

The current piece of work reflects an interesting idea as a Fuzzy based Efficient Healthcare Data Collection and Analysis Mechanism using Edge Nodes in IoMT. I have found many issues which convinced me to recommend a major revision.

1.            Challenges and Motivation is quite Clear: “Internet of Things (IoT) is an advanced technology that comprises of enormous devices having sensors to collect, send and receive data. Due to its vast popularity and efficiency, it is employed in collecting crucial data of health sector. As the sensors generate huge amount of data, it is better to be aggregated before transmitting it further. These sensors generate data frequently so when there is no variation in it, same data is produced. The base scheme has no mechanism to comprehend duplicate data. This problem has negative effect on performance of heterogeneous network as it increases energy consumption and high control overhead. To overcome these issues, this paper presents.” However, I will highly recommend adding a one-line methodology to make the abstract more complete.

2.            Also add a brief description of methodology in introduction section.

3.            Revise the entire work for ambiguous sentences, i.e., Line 101 to 296.

4.            Provide comparison based on various frame of reference in discussion section.

5.  explain Equation 4.

6. In Algorithm 1 Parent Selection at Control Phase, what is the functionality of line 17 to 23?

Author Response

Original Manuscript ID: sensors-2470844

Original Article Title: “Fuzzy based Efficient Healthcare Data Collection and Analysis Mechanism using Edge Nodes in IoMT

To: Sensor Editor

Re: Response to reviewers

Dear Editor,

Thank you for allowing a re-submission of our manuscript, with an opportunity to address the reviewers’ comments.

We are uploading (a) our point-by-point response to the comments, (b) an updated manuscript by highlighting the modified content, and (c) a clean updated manuscript without highlights.

Best regards,

<Muhammad Nafees Ulfat khan> et al.

Reviewer 2 Report

Review of the paper Fuzzy based Efficient Healthcare Data Collection and Analysis Mechanism using Edge Nodes in IoMT, published in MDPI Sensors.

The paper entitled Fuzzy based Efficient Healthcare Data Collection and Analysis Mechanism using Edge Nodes in IoMT presents a system that applies fuzzy logic combined with Boolean duplicated values to reduce energy consumption in heterogeneous networks.

The paper should be improved in the following aspects:

1.     The English needs to be improved. There are many typing errors throughout the text.

2.     Section 4.1 is unclear and needs to be rewritten.  There should be a block diagram at the beginning of the section showing the different functions or algorithms proposed in your system to give the reader a general idea of the relationship between functions.

3.     The units of the axes in the figures in Section 4 are not explained. Are they expressed as percentages? Describe it in the text.

4.     Line 6 of Algorithm 1, “w denotes source”. Correct the typo.

5.     From line 26 Algorithm 1 to the end, is not an algorithm, it is a description combined with formulas.

6.     Table 2 is not referenced in the text.

7.     Section 5. Why do you choose NS-2.35 instead of NS-3?

8.     Line 454: “The nodes are placed in these grids with probability”. What distribution do you use to place the nodes in the grid? Explain in the text.

9.     How is the data from the sensors generated, from a database, statistically…?

10.  How many replicates of each experiment are carried out?

11. Why do the results only show the mean, but not other statistical information such as confidence intervals, variance, ...?

12.  DICA_EXTENSION line exchange colour with DICA in some figures.

The English needs to be improved. There are many typing errors throughout the text, and sentences that are not understood.

Author Response

(The authors gave the same response as above.)

Round 2

Reviewer 2 Report

The comments from the previous review have been addressed, which has improved the paper. However, there are still minor typos as, for example, line 383: "is shown in the blcok diagram...". Please, check the text again.

Regarding comment 12 of the previous review, you state that "the confidence intervals may be shown for these graphs", but they are not displayed. Please, plot the confidence intervals, which provides additional information in order to properly evaluate the results. 

The quality of the English has improved, but there are still minor typos.

Author Response

Original Manuscript ID: sensors-2470844

Original Article Title: “Fuzzy based Efficient Healthcare Data Collection and Analysis Mechanism using Edge Nodes in IoMT

To: Sensor Editor

Re: Response to reviewers

Dear Editor,

Thank you for allowing a re-submission of our manuscript, with an opportunity to address the reviewers comments.

We are uploading (a) our point-by-point response to the comments, (b) an updated manuscript by highlighting the modified content, and (c) a clean updated manuscript without highlights.

Best Regards,

<Muhammad Nafees Ulfat khan> et al.
